



# Development of a regionally consistent and fully probabilistic earthquake risk model for Central Asia

Mario A. Salgado-Gálvez[1], Mario Ordaz[1,2], Benjamín Huerta[1], Osvaldo Garay[1], Carlos Avelar[1], Ettore Fagà[3], Mohsen Kohrangi[3], Paola Ceresa[3], Zacharias Fasoulakis[3]

[1] ERN International. Mexico City, Mexico.
[2] Instituto de Ingeniería, Universidad Nacional Autónoma de México. Mexico City, Mexico
[3] RED Risk Engineering Development, Pavia, Italy

*Correspondence to*: Mario A. Salgado-Gálvez (mario.sal.gal@gmail.com)

**Abstract**

A fully probabilistic earthquake risk model was developed for five countries in Central Asia, providing updated earthquake loss estimates with a higher level of details on all components with respect to previous studies in the region, besides having used a regionally consistent approach that on the one hand, allows direct comparisons at different disaggregation levels (e.g., Country and Oblast), and on the other hand, is aimed to facilitate initiating a policy dialogue regarding national and regional disaster risk financing and insurance applications. This earthquake risk model made use of a regional probabilistic seismic hazard analysis, as well as a comprehensive exposure database on which different types of assets and sectors were included, and for which two scenarios (years 2020 and 2080) were modelled. For each type of asset, a unique vulnerability function was derived and later used for the convolution with the hazard data that allowed estimating the loss exceedance curve, at different disaggregation levels, from where other risk metrics such as the average annual loss (AAL) and specific return period losses, were obtained. The regional earthquake AAL for the 2020 exposure scenario has been estimated in around $2 Bn, being Kyrgyzstan and Tajikistan the countries with the highest earthquake risk levels in the region. Besides the probabilistic earthquake risk results, as-if scenarios were modelled using a pseudo-deterministic approach to assess the human and economic losses for realistic and representative earthquakes for the main cities within earthquake prone regions in the five countries within the study area.

**Short summary**

Central Asia is prone to earthquake losses which can impact population and assets of different types. This paper presents the details of a probabilistic earthquake model which made use of a regionally consistent approach to assess the feasible earthquake losses in five countries. Results are presented in terms of commonly used risk metrics, which are aimed to facilitate a policy dialogue regarding different disaster risk management strategies, from risk mitigation to disaster risk financing.

## 1 Introduction and Previous Studies

A regionally consistent and fully probabilistic earthquake risk model was developed for Central Asia, under the sponsorship of the World Bank, in the framework of the *Strengthening Financial Resilience and Accelerating Risk Reduction in Central Asia* project (SFRARR). Central Asia is an area characterized by a complex and active tectonic deformation, on which earthquakes besides having the possibility to inflict damages and losses because of the ground motion, can trigger secondary hazards such as landslides, and naturally dammed lake outbursts. The model described in this paper provides updated earthquake loss results for Kazakhstan, Tajikistan, Turkmenistan, Uzbekistan, and the Kyrgyz Republic. To achieve this, it was needed the development of a probabilistic seismic hazard analysis (PSHA), an exposure database for buildings and infrastructure (representative of years 2020 and 2080), and a set of earthquake vulnerability functions for the representative building classes and infrastructure assets.

Within this same project, a probabilistic flood risk assessment was carried out for the same exposed assets, which details can be found in Coccia et al., (2023). The project required a multi-hazard risk assessment, hence a peril-agnostic risk assessment methodology was chosen, which is based on the proposal by Ordaz (2000) which has been implemented in the R-CAPRA software (ERN, 2022), a tool that has been used for probabilistic risk assessments for different hazards and at different





resolution levels (see for instance Salgado-Gálvez et al., 2014; Jaimes et al., 2016; Ordaz et al., 2019). Earthquake and flood hazards are represented through synthetic catalogs of 10000 years, whereas the relationship between the hazard intensity measures (i.e., ground motion for earthquakes and water depth for floods) and the expected losses (both human and economic) is represented through vulnerability functions.

There have been previous regional and national earthquake risk assessments in Central Asia, such as the one developed in 2009 by the Central Asia and Caucasus Disaster Risk Management Initiative (CAC-DRMI, 2009), which generated earthquake risk profiles at national, regional, and sub-regional levels, based on historical data covering the 1988-2007 period. In the framework of the Global Risk Model by the United Nations Office for Disaster Risk Reduction (UNDRR), a global and probabilistic earthquake risk assessment was carried out for 216 countries, which included the five nations that are part of this

study, obtaining reference values in terms of average annual losses (AAL) and loss exceedance curves (LEC) for earthquakes and floods (Ordaz et al., 2014; Cardona et al., 2014; UNISDR, 2015). In 2016, The Global Facility for Disaster Reduction and Recovery (GFDRR) produced earthquake risk profiles for Central Asia at provincial and national levels, with results in terms of affected people and economic losses for multiple return periods, the latter normalized by the country gross domestic product (GDP) (GFDRR, 2016). More recently, in 2017, an earthquake risk assessment was carried out for the Kyrgyz Republic (World

Bank, 2017) which estimated economic losses for different sectors (e.g., residential buildings, schools, fire stations, roads, bridges, and hospitals), although loss metrics such as the AALs are not available for all of them; the same study provided also human and economic losses for 12 scenarios derived using a deterministic methodology. Some studies were developed for the region also to quantify earthquake risk at urban scale, like the one carried out by the Government of Japan which analyzed three major historical earthquakes that caused significant losses and disruptions in Almaty, with the aim of providing the bases

for the formulation of the earthquake disaster risk management plan for Almaty City (JICA, 2009).

The earthquake model described in this paper was developed with the objective of facilitating strategic discussions with relevant stakeholders, allowing for coherent and consistent strategic financial solutions across the geographical study area and the key economic sectors of the five countries, as well as to inform the World Bank's engagement in supporting regional and

national disaster risk financing and insurance applications, as for instance traditional and parametric solutions for the structuring of a regional program. However, results are only intended to inform and enable the World Bank to initiate a policy dialogue, and do not have yet the sufficient detail to recommend, or support planning the design of specific disaster risk management initiatives. The results of the earthquake risk assessment include LECs and year loss tables (YLT), that can be further disaggregated at two administrative levels (country – ADM0, and Oblast – ADM1). In addition, the model allows

obtaining return period loss estimates and AALs for these two aggregation levels, and for each sector included in the exposure model.

Four (4) different exposure models were developed in this project (Scaini et al., 2023a; 2023b), where the first one corresponds to a reliable and detailed representation of the conditions for year 2020 for multiple sectors, whereas the other three depict a

projection to year 2080, for the residential sector only, considering three different shared socioeconomic paths (SSP), namely SSP1, SSP4 and SSP5. Since earthquake hazard can be assumed as stationary, the occurrence characteristics of the PSHA developed for todays' conditions can be considered as the same for the next 60 years, reason why no variations in the hazard model for the year 2080 earthquake risk estimates were introduced.

For year 2020 exposure, the regional AAL has been estimated in approximately $2 Bn, finding that the countries with the highest relative losses (i.e., national earthquake losses normalized by the country's exposed value) are the Kyrgyz Republic and Tajikistan, with results over 3‰, whereas those with the lowest earthquake risk are Kazakhstan and Turkmenistan. For



the projected exposure to year 2080, and considering that it is only representative for the residential buildings, the regional AAL has been estimated in approximately $0.9 Bn, again finding that the Kyrgyz Republic and Tajikistan are the countries

with the highest expected earthquake risk in the region in the future.

In addition to the probabilistic risk results, a pseudo-deterministic method was used to estimate the human and economic losses for five credible and feasible earthquakes, one for each capital city (except for Kazakhstan where the scenario analysis was carried out for Almaty City). These earthquakes were selected from the YLT after having disaggregated the losses for a 100-

year return period. Finally, the modelled losses have been used to derive a relationship between them and the total emergency costs, which details can be found in Berny et al., (2023), to complement the relevant information required for the design and implementation of a comprehensive disaster risk management plan.

## 2 Risk Assessment Methodology

This section describes the fully probabilistic risk assessment methodology that has been used to estimate the potential

earthquake losses in Central Asia. As previously mentioned, it is peril-agnostic and is the same that was used for estimating flood losses in the same study area, which details and results can be found in Coccia et al., (2023). The main objective of any probabilistic risk assessment, regardless of the hazard, is to provide a long-term relationship between the losses (e.g., fatalities and/or economic losses), and t

heir occurrence frequencies. Figure 1 shows the general framework of the risk assessment methodology used in this study,

noting that it is at the loss module where the combination of the different outputs of the hazard, exposure, and vulnerability modules is made, yielding in this case the estimates of human and economic losses induced by earthquakes.

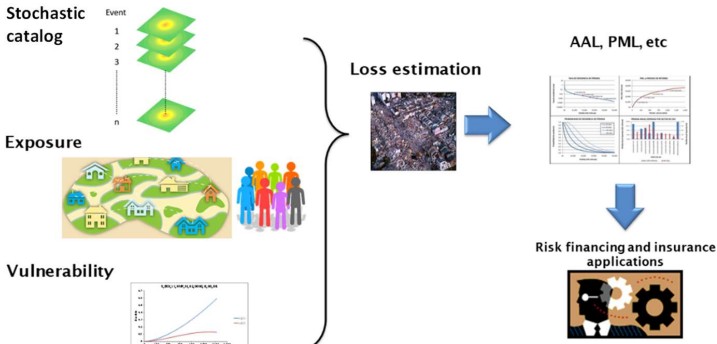

**Figure 1: Components and results for the risk assessment**

The probabilistic risk assessment methodology employed in this study required the following analytical steps:

• A probabilistic seismic hazard analysis (PSHA) which output consists of a synthetic earthquake catalog with a duration of 10,000 years, which was deemed as a long enough timespan that provided an acceptable balance between risk results stability (for long return periods) and the computational effort. This synthetic earthquake catalog contains thousands of earthquake events, for which on each case, the characteristics about the location, depth, magnitude, and geometry of the rupture (e.g., strike, dip, shape, aspect ratio) are included, so that the estimation of the probability

distribution of the ground motion intensities (i.e., ground acceleration) produced by the event in the surrounding region can be made. For generating the synthetic earthquake catalog, a PSHA was developed for the area under study, which details are described in Poggi et al., (2023a; 2023b) and outputs were converted into the format required by R-CRISIS (Ordaz et al., 2021; 2023) so that it could be used in R-CAPRA, the risk assessment tool.


- Definition of the inventory of exposed assets: for the five countries in the study area of this project, different exposure models (1 for year 2020 and 3 for year 2090) were developed including information about the location of the assets, their replacement cost, and their structural characteristics (e.g., construction material, height, structural system, among others). The exposure module for year 2020 accounts for the following types of assets:

- Population
- Building stock
  - Residential buildings
  - Non-residential buildings (schools, healthcare facilities, industrial and commercial buildings)
- Infrastructure
  - Transportation system (roads, railways, and bridges)
  - Airports and airstrips
  - Supply infrastructure

Each one of the other three exposure models that provide a projection of the exposure for year 2080 considered a different Shared Socioeconomic Path (SSP): SSP1, SSP4 and SSP5, and only the residential buildings have been
included. The full details of the development of the complete set of exposure models can be found in Scaini et al., (2023a; 2023b).

- The relationship between the hazard intensity measure and the expected human and physical losses was represented through vulnerability functions, which provide a continuous, quantitative, and probabilistic relationship between the
hazard intensity measure (ground acceleration in this case), the expected loss, and a dispersion measure. A unique vulnerability function was developed for each class of asset included in the exposure datasets (buildings and different types of infrastructure).

- Hazard, exposure, and vulnerability data were combined in the loss module, where for each synthetic earthquake, the
possible economic and human losses were estimated for the five countries. In a nutshell, the human and economic losses for each exposed asset were computed after convolving the ground acceleration distribution at the site of interest, with the corresponding vulnerability function. This procedure provides a distribution of the mean loss ratio (i.e., the repair cost normalized by the asset replacement cost), which is later multiplied by the total value of the asset to obtain the distribution of losses for the asset caused by a given earthquake. The total loss for each synthetic
earthquake is obtained after summing up the losses for all exposed assets. Because every synthetic earthquake has an annual occurrence probability, the losses for all events in the catalog are combined using the methodology proposed by Ordaz (2000), which is explained in detail next.

The probability density function of the loss for the $i^{th}$ event is computed after aggregating the losses of the individual assets in the exposure database. The expected value of the loss, denoted as $E(l|Event_i)$, and its variance, $\sigma^2(l|Event_i)$ is calculated, for
each earthquake in the synthetic catalog, with the following expressions:

$$E(l|Event_i) = \sum_{j=1}^{NE} E(l_j) \tag{1}$$

$$\sigma^2(l|Event_i) = \sum_{j=1}^{NE} \sigma^2(l_j) + 2\sum_{\substack{k=1 \\ k<j}}^{NE-1} \sum_{j=2}^{NE} cov(l_k, l_j) \tag{2}$$



where *NE* represents the total number of exposed assets within the ground motion footprint, $E(l_j)$ the expected value of the loss of the $j^{th}$ exposed asset given the occurrence of the $i^{th}$ event, $\sigma^2(l_j)$ corresponds to the variance of the loss at the $j^{th}$ exposed asset given the occurrence of the $i^{th}$ event, and $cov(l_k,l_j)$ the covariance of the loss of two different exposed assets. This covariance is estimated by using a correlation coefficient, denoted as $\rho_{k,j}$, besides considering the standard deviations of the losses for different exposed assets. Equation 2 can therefore be rewritten as:

$$\sigma^2(l|Event_i) = \sum_{j=1}^{NE} \sigma^2(l_j) + 2 \sum_{\substack{k=1 \\ k<j}}^{NE-1} \sum_{j=2}^{NE} \rho_{k,j}\sigma(l_k)\sigma(l_j) \tag{3}$$

The key outcome of this fully probabilistic risk assessment methodology corresponds to the LEC, which provides a relationship between different loss values and their exceedance frequencies. This model has calculated the LEC using the following expression, which corresponds to one of the possible ways that the Total Probability Theorem can adopt.

$$v(l) = \sum_{i=1}^{N} \Pr(L > l|Event_i) \cdot F_A(Event_i) \tag{4}$$

*v(l)* represents the exceedance rate of the loss *l*, $\Pr(L>l|Event_i)$ is the probability that the loss is larger than *l* given the occurrence of the $i^{th}$ event, and $F_A(Event_i)$ is the annual occurrence frequency of the $i^{th}$ event. The sum of the equation is carried for all the earthquakes included in the synthetic catalog. Finally, the return period of any loss value of interest, *Tr(l)*, can be calculated as the inverse value of its loss exceedance rate.

$$Tr(l) = \frac{1}{v(l)} \tag{5}$$

Since the loss computed in a group of exposed assets during a given synthetic earthquake is an uncertain quantity, it must be treated as a random variable. We have considered the uncertainty in the occurrence of future earthquakes, the uncertainty related to the estimation of the hazard intensities caused by an event (i.e., through the σ value of the ground motion models, GMMs) and the uncertainty in the vulnerability functions (i.e., the dispersion measure for the expected loss).

The loss probability distribution for each synthetic earthquake is calculated by first determining the distribution of the hazard intensity value at the location of each exposed asset and then evaluating the probability distribution of the loss given that hazard intensity value. This is a standard approach that simplifies the problem of assessing the loss distribution by dividing it into two steps. The probability of exceeding a loss with a given value *l*, conditioned to the occurrence of an earthquake, is thus expressed as follows:

$$\Pr(L > l|Event_i) = \int_I \Pr(L > l|I)\, f(I|Event_i)\, dI \tag{6}$$

In Equation 6, $\Pr(L>l|I)$, corresponds to the probability that the loss will exceed the value *l* given that the local ground motion intensity was *I*. This term represents the vulnerability model and the uncertainty associated with the expected loss given a certain value of ground motion. $f(I|Event_i)$ represents the probability density function of the hazard intensity, conditional to the occurrence of the event. In this case, it represents the evaluation of the ground motion, which is uncertain, given the occurrence of an earthquake by means of the GMMs. The probability distribution of the ground motion intensity includes the aleatory uncertainty associated with the estimate of the ground motion caused by the event as computed by a GMM and the epistemic uncertainty due to the use of more than one GMM to assess the ground motion intensity (i.e., logic-trees).





## 3 Earthquake vulnerability functions and loss validation and calibration

### 3.1 Development of a set of regional earthquake vulnerability functions

For each type of asset included in the exposure database, a unique earthquake vulnerability function was derived to estimate
the human and economic losses using ground acceleration as hazard intensity measure. Vulnerability functions can be derived
using different approaches, namely analytical, empirical, and expert opinion based, although usually a combination of these is
used. In this project, the empirical approach (i.e., the one that uses post-earthquake damage and loss observed data for different
types of assets) was discarded because of the lack of sufficient data about ground motion and building damages and losses
from past earthquakes occurred in the study area. The methodology adopted involved collecting and reviewing multiple sets
of existing fragility and vulnerability functions from various references relevant for the region, including local studies,
international literature and World Bank projects previously developed in the region. Such analysis allowed to create a large
database of vulnerability functions that were classified in accordance with the taxonomy used in our exposure model. These
functions were further harmonized and processed, and a single class-specific function was extracted to capture the mean and
variation in the database compiled.

.

Because of the objective of the project and the lack of local data, in particular for some countries in the region, the earthquake
vulnerability functions were derived at regional level considering the similarities in construction practice in the whole area.
Hence no differences between the countries could be captured, although they may exist. As explained in Scaini et al., (2023a;
2023b), six building types and fifteen building subtypes were included in the taxonomy for the residential Central Asian
buildings. For the non-residential building category eight building types were identified whereas for the infrastructure assets
(e.g., roads, railways, and bridges), 10 classes were included.

The starting point for the development of the regional vulnerability model was to compile a set of fragility curves (Kircher and
McCann, 1983), which define the probability of exceeding a damage state (DS) given the ground motion intensity, properly
classified in accordance to the exposure taxonomy. These are later combined with consequence functions that define the
expected loss ratio (i.e., the fraction of earthquake losses with respect to the total replacement cost of an asset), for a given DS.
To define these parameters, different studies previously developed for the region were reviewed, as, for instance, the works by
Kostov et al., (2004); Ahmad et al., (2011); Karantoni et al., (2011); and Lagomarsino and Cattari (2014), as well as the existing
national earthquake resistant building codes and the Seismic Risk Assessment in the Kyrgyz Republic by the World Bank
(2017). Also, for specific buildings such as schools, global libraries of vulnerability functions such as the one by the Global
Library of School Infrastructure (GLOSI) were considered. The functions collected were then harmonized and processed to
derive a unique function for each building class, accounting for the uncertainty implied by the combination of functions derived
with different approaches and methodologies. This allowed, for the first time in Central Asia context, deriving a new model
applicable for the entire region leveraging the most recent international research outcomes and the local observations and
expertise. Moreover, the approach adopted allowed to consider the uncertainty associated with the evaluation of the damage
due to the different approaches that could be adopted to define the vulnerability (i.e., the variability among the studies
analyzed).

Figure 2 shows some examples of the earthquake vulnerability functions used in this study to estimate earthquake losses on
different types of buildings and infrastructure components and full details of the vulnerability models can be found in RED
(2023).

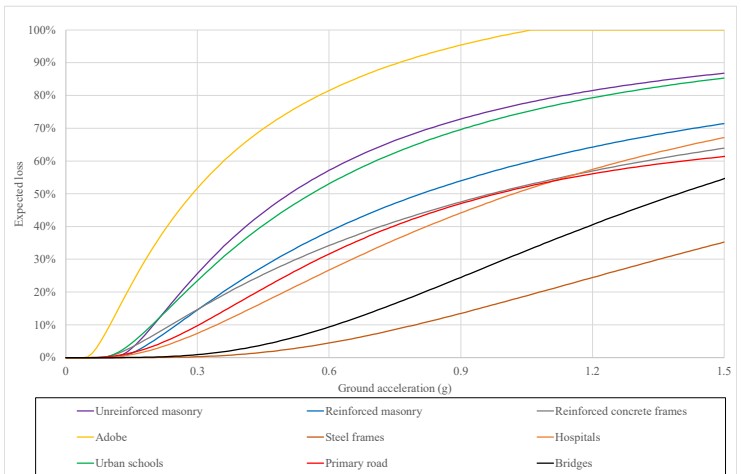

**Figure 2: Example of earthquake vulnerability functions for Central Asian residential buildings and infrastructure assets**

### 3.2 Loss model calibration and validation

A calibration procedure was carried out for the earthquake vulnerability model making use of data for 7 historical earthquakes in the region, for which observed (reported) losses were available. This was an iterative process on which the original vulnerability functions overestimated the modeled losses and therefore, it was decided to include adjustments. More specifically, modeled earthquake losses were validated making use of the publicly available data of historical earthquakes in the study area. This process was carried out by comparing the results of this model on an event-by-event basis, in terms of economic losses and fatalities, with those included in available reports. Reported losses were obtained from different sources and it must be highlighted that these values are affected by large uncertainties and the following limitations exist:

- The accuracy of the reported losses is variable since different methodologies were used, at different times, in each of the five countries. Because of the use of different methodologies, the same event could have different values of reported losses.

- The reported losses were trended using the methodology by Pielke et al., (2003) to account for the effect of the national population, inflation, local currency deflator and gross domestic product growth, among others to make them comparable with the modeled values. This process is another source of uncertainty because many factors that occurred after the event are unknown. In general, the oldest the event, the most uncertain its reported losses.

- Reported losses usually account for the sum of direct and indirect (and in some cases content) losses and this detail is often unspecified, or the losses are not disaggregated by type. The earthquake losses computed in this study are representative only for the direct losses on buildings and infrastructure.

- For earthquakes such as the 2003 Kazakhstan event, the available economic losses were reported the local currency (Tenge). Since the comparisons were carried out in terms of US dollars, additional variability was included to the procedure by adopting an average exchange rate for the time of occurrence.

With the support of the local specialists involved in the development of this project, data from the seven historical earthquakes listed in Table 1 were compiled and used for validation purposes.

**Table 1: Location, magnitude and depth parameters for the historical earthquakes used in the loss validation process**




| Event ID | Country | Date | Mw | Latitude (°) | Longitude (°) | Depth (km) |
|---|---|---|---|---|---|---|
| 1 | KGZ | 05/10/2008 | 6.7 | 73.44 | 39.31 | 40 |
| 2 | KGZ | 19/08/1992 | 7.3 | 73.63 | 42.07 | 25 |
| 3 | UZB | 20/07/2011 | 6.3 | 71.42 | 40.16 | 20 |
| 4 | UZB | 26/05/2013 | 6.2 | 67.40 | 39.20 | 18 |
| 5 | TJK | 07/12/2015 | 7.2 | 72.78 | 38.21 | 22 |
| 6 | TJK | 29/07/2006 | 5.6 | 68.83 | 37.26 | 34 |
| 7 | KAZ | 23/05/2003 | 6 | 80.52 | 42.91 | 10 |

With the data shown in Table 1 and considering the GMM and rupture's characteristics that were used in the PSHA for this project, the acceleration footprints were generated to estimate the losses for the different types of assets. The methodology employed for calculating the losses of these historical earthquakes is the same as for the fully probabilistic risk assessment. For most of the historical earthquakes, there are reported two possible fault planes from the moment tensor solution and then, based on the tectonic environment while maintaining coherence with the PSHA assumptions, the most appropriate was chosen

to define the geometric and orientation characteristics of each rupture. A similar methodology was followed for the calibration of the vulnerability functions to estimate human losses, for which a comparison between the modeled and reported fatalities was made, allowing improving the fit between the two values. Figure 3 shows the comparison between the reported and modeled losses (before and after the calibration).

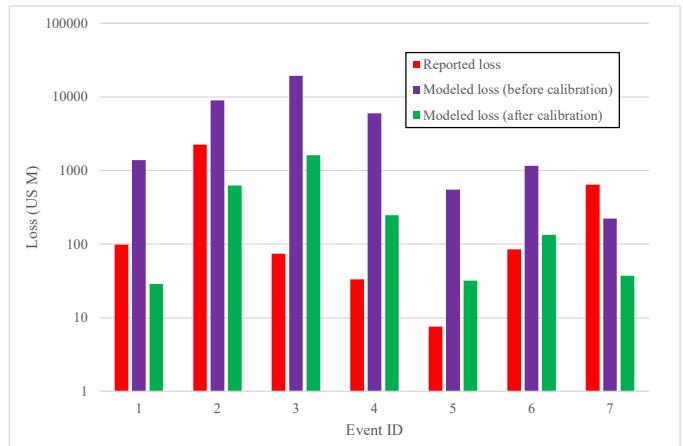


**Figure 3: Comparison of monetary losses between the observed (reported) values and the modeled losses before and after the calibration**

Figure 4 shows as example, the shakemaps, for modelled peak ground acceleration (PGA) for the Kyrgyz Republic earthquakes listed in Table 1. Figure 5 shows the comparison between the modelled and reported economic losses and fatalities for the two

events. The average value is indicated in green and is limited by the black squares that depict the upper and lower limits of the reported values, as per the different data sources.

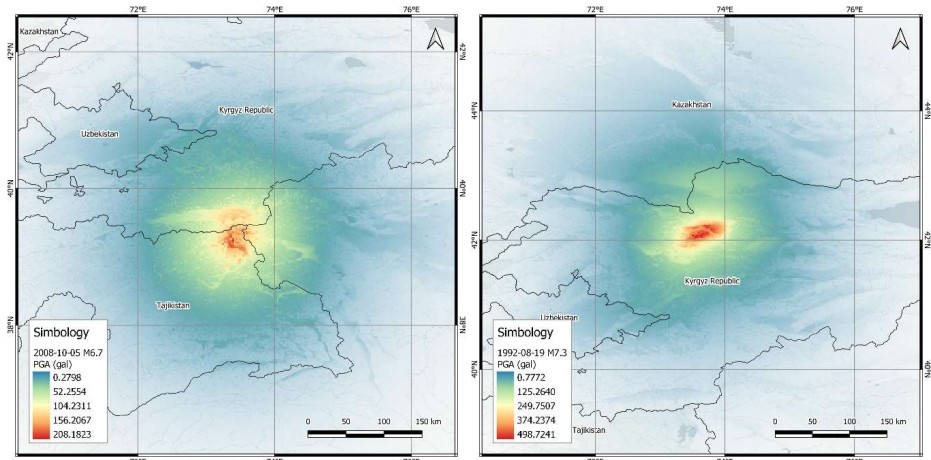

**Figure 4: Shakemaps (PGA) for the October 5th, 2008 (left) and the August 19th, 1992 (right) Kyrgyzstan earthquakes**

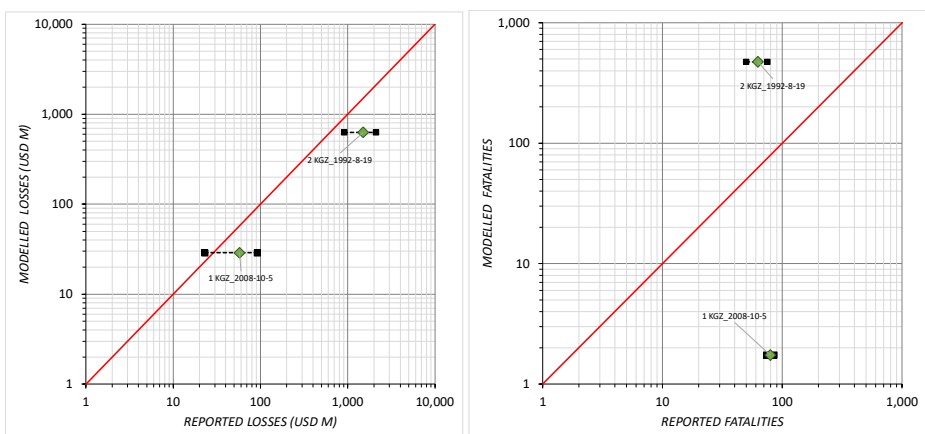

**Figure 5: Comparison of modelled economic losses (left) and fatalities (right) for the historical earthquakes in the Kyrgyz Republic**

The reported economic losses for the two historical earthquakes used in the validation procedure in the Kyrgyz Republic are higher than the modelled losses by a factor of almost three. However, it should be stressed again that the reported losses have large uncertainties associated, as for instance what they include (e.g., only residential buildings? Emergency costs?). For the modelled fatalities, of the 2008 event the obtained results are lower than the reported one, whereas the opposite occurs for the 1992 event. As mentioned before, the use of a regional vulnerability model can leave aside different considerations that are particular for a country and therefore, these differences can be explained by that.

The differences between the modelled and the reported losses for the 7 historical events, despite being non-negligible can be explained as follows. First, during the development of the earthquake vulnerability functions for this project there were not enough specific data, neither about the vulnerability characteristics, nor about the reported losses to be able to distinguish and calibrate the models at country level. Therefore, a regional approach was adopted for the development of the vulnerability functions. As shown in Figure 2, there are some overestimations (e.g., Uzbekistan 2011 earthquake) and underestimations (e.g., Kyrgyzstan 1992 earthquake) against the reported losses that are normal in this type of modelling, even if these are systematic for a given country. Even if part of the differences can be related to the characteristics of the exposed assets at the time of the event, compared to the current ones, another possible explanation could be the choice of conservative (high) loss ratios, particularly for low acceleration levels, which can affect the aggregated losses. This occurs because the number of





exposed assets in areas far from the epicenter (and thus affected by low ground motion intensities) is usually much larger than that close to it for these 7 events, and even very small loss ratios in the low acceleration range, when summed for all those far exposed assets, might unrealistically increase the total aggregated losses. The other components of the risk assessment

(earthquake hazard and exposure) were subject to separate validation processes which details can be found in Poggi et al., (2023a; 2023b) and Scaini et al., (2023a; 2023b).

## 4 Results

### 4.1 Aggregated earthquake loss estimates

Earthquake risk results have been calculated at different aggregation levels, being the Oblast one the most refined, followed

with the national and by the regional cases. In the latter, aggregated results for the five countries in the study area are shown. For each case, a LEC has been estimated, as shown in Figure 6, providing a relationship between different loss levels and their annual occurrence frequencies. As it is well-known, the inverse value of the annual occurrence frequency represents the return period (in years).

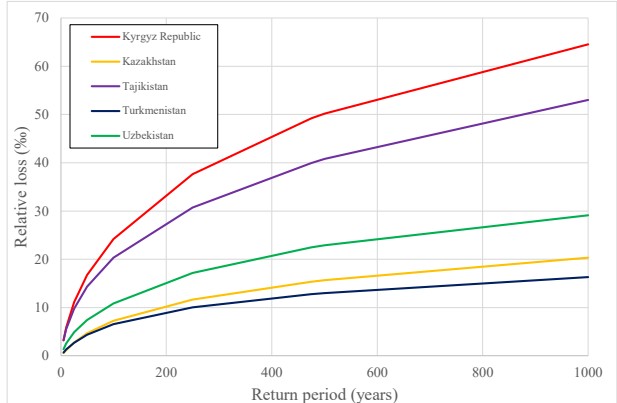


**Figure 6: Country level earthquake loss exceedance curves**

Tables 2 to 5 show the tabulated results for the earthquake losses (in absolute values and normalized by the exposed value) in terms of the AAL and return period losses between 5 and 1000 years at country and regional level for the 2020 exposure and the three SSPs for year 2080. Appendix 1 includes the earthquake risk results at Oblast level, for the residential sector and year

2020 exposure for the five countries in the study area.

**Table 2. Tabulated earthquake losses at regional and country level for the 2020 exposure (all sectors)**

| Tr (years) | Absolute values (US M) | | | | | | Relative values (‰) | | | | | |
|---|---|---|---|---|---|---|---|---|---|---|---|---|
| | Regional | KGZ | KAZ | TJK | TKM | UZB | Regional | KGZ | KAZ | TJK | TKM | UZB |
| 5 | $1,797 | $184 | $303 | $247 | $38 | $1,280 | 1.07 | 3.18 | 0.54 | 3.27 | 0.68 | 1.38 |
| 10 | $3,031 | $335 | $648 | $416 | $72 | $2,380 | 1.81 | 5.8 | 1.15 | 5.51 | 1.29 | 2.57 |
| 25 | $5,403 | $640 | $1,549 | $738 | $149 | $4,548 | 3.22 | 11.1 | 2.76 | 9.78 | 2.7 | 4.91 |
| 50 | $7,929 | $964 | $2,629 | $1,082 | $242 | $6,873 | 4.73 | 16.7 | 4.68 | 14.3 | 4.38 | 7.41 |
| 100 | $11,331 | $1,397 | $4,066 | $1,533 | $362 | $10,065 | 6.75 | 24.2 | 7.23 | 20.3 | 6.54 | 10.9 |
| 250 | $17,366 | $2,174 | $6,528 | $2,320 | $554 | $15,913 | 10.35 | 37.7 | 11.6 | 30.7 | 10 | 17.2 |
| 475 | $22,518 | $2,841 | $8,627 | $3,018 | $708 | $20,842 | 13.42 | 49.2 | 15.4 | 40 | 12.8 | 22.5 |
| 500 | $22,955 | $2,898 | $8,807 | $3,079 | $720 | $21,252 | 13.68 | 50.2 | 15.7 | 40.8 | 13 | 22.9 |
| 1000 | $29,175 | $3,724 | $11,426 | $4,000 | $902 | $26,971 | 17.39 | 64.5 | 20.3 | 53 | 16.3 | 29.1 |
| **AAL** | **$1,924** | **$192** | **$351** | **$238** | **$34** | **$1,109** | **1.15** | **3.32** | **0.62** | **3.15** | **0.62** | **1.2** |





Table 3. Tabulated earthquake losses at regional and country level for the 2080 exposure (residential sector only), SSP1

| Tr (years) | Absolute values (US M) | | | | | | Relative values (‰) | | | | | |
|---|---|---|---|---|---|---|---|---|---|---|---|---|
| | Regional | KGZ | KAZ | TJK | TKM | UZB | Regional | KGZ | KAZ | TJK | TKM | UZB |
| 5 | $814 | $95 | $131 | $166 | $14 | $549 | 0.72 | 2.96 | 0.39 | 2.97 | 0.70 | 0.80 |
| 10 | $1,389 | $161 | $275 | $270 | $26 | $1,043 | 1.23 | 5.00 | 0.82 | 4.84 | 1.31 | 1.52 |
| 25 | $2,530 | $291 | $668 | $464 | $53 | $2,038 | 2.24 | 9.05 | 1.98 | 8.32 | 2.64 | 2.97 |
| 50 | $3,772 | $433 | $1,211 | $667 | $84 | $3,121 | 3.33 | 13.44 | 3.60 | 11.96 | 4.21 | 4.55 |
| 100 | $5,472 | $631 | $1,994 | $936 | $126 | $4,618 | 4.84 | 19.58 | 5.92 | 16.78 | 6.31 | 6.73 |
| 250 | $8,609 | $1,011 | $3,452 | $1,427 | $196 | $7,510 | 7.61 | 31.40 | 10.26 | 25.59 | 9.87 | 10.94 |
| 475 | $11,361 | $1,362 | $4,793 | $1,885 | $256 | $10,150 | 10.04 | 42.29 | 14.24 | 33.80 | 12.87 | 14.78 |
| 500 | $11,597 | $1,393 | $4,912 | $1,926 | $261 | $10,378 | 10.25 | 43.25 | 14.59 | 34.53 | 13.12 | 15.11 |
| 1000 | $15,017 | $1,846 | $6,681 | $2,550 | $335 | $13,603 | 13.28 | 57.33 | 19.85 | 45.73 | 16.81 | 19.81 |
| **AAL** | **$931** | **$103** | **$163** | **$162** | **$13** | **$491** | **0.82** | **3.19** | **0.48** | **2.9** | **0.63** | **0.72** |

Table 4. Tabulated earthquake losses at regional and country level for the 2080 exposure (residential sector only), SSP4

| Tr (years) | Absolute values (US M) | | | | | | Relative values (‰) | | | | | |
|---|---|---|---|---|---|---|---|---|---|---|---|---|
| | Regional | KGZ | KAZ | TJK | TKM | UZB | Regional | KGZ | KAZ | TJK | TKM | UZB |
| 5 | $822 | $96 | $131 | $166 | $14 | $557 | 0.73 | 2.97 | 0.39 | 2.96 | 0.70 | 0.81 |
| 10 | $1,403 | $162 | $275 | $271 | $26 | $1,056 | 1.24 | 5.02 | 0.82 | 4.83 | 1.30 | 1.54 |
| 25 | $2,553 | $293 | $668 | $467 | $53 | $2,062 | 2.25 | 9.08 | 1.98 | 8.32 | 2.64 | 3.00 |
| 50 | $3,802 | $435 | $1,211 | $673 | $84 | $3,153 | 3.36 | 13.49 | 3.60 | 11.98 | 4.22 | 4.58 |
| 100 | $5,511 | $634 | $1,996 | $946 | $126 | $4,661 | 4.87 | 19.63 | 5.93 | 16.82 | 6.34 | 6.78 |
| 250 | $8,668 | $1,014 | $3,455 | $1,443 | $197 | $7,572 | 7.65 | 31.42 | 10.27 | 25.67 | 9.91 | 11.01 |
| 475 | $11,436 | $1,364 | $4,795 | $1,906 | $257 | $10,231 | 10.10 | 42.27 | 14.26 | 33.91 | 12.92 | 14.87 |
| 500 | $11,676 | $1,395 | $4,914 | $1,947 | $262 | $10,459 | 10.31 | 43.22 | 14.61 | 34.64 | 13.17 | 15.20 |
| 1000 | $15,120 | $1,848 | $6,683 | $2,577 | $336 | $13,707 | 13.35 | 57.29 | 19.87 | 45.86 | 16.87 | 19.93 |
| **AAL** | **$938** | **$103** | **$163** | **$163** | **$13** | **$497** | **0.83** | **3.2** | **0.48** | **2.89** | **0.63** | **0.72** |

Table 5. Tabulated earthquake losses at regional and country level for the 2080 exposure (residential sector only), SSP5

| Tr (years) | Absolute values (US M) | | | | | | Relative values (‰) | | | | | |
|---|---|---|---|---|---|---|---|---|---|---|---|---|
| | Regional | KGZ | KAZ | TJK | TKM | UZB | Regional | KGZ | KAZ | TJK | TKM | UZB |
| 5 | $822 | $96 | $132 | $166 | $14 | $557 | 0.73 | 2.98 | 0.39 | 2.99 | 0.71 | 0.81 |
| 10 | $1,404 | $163 | $276 | $269 | $26 | $1,057 | 1.24 | 5.05 | 0.82 | 4.86 | 1.31 | 1.54 |
| 25 | $2,557 | $296 | $670 | $463 | $53 | $2,065 | 2.26 | 9.15 | 1.99 | 8.34 | 2.65 | 3.00 |
| 50 | $3,808 | $440 | $1,216 | $664 | $84 | $3,159 | 3.36 | 13.60 | 3.61 | 11.97 | 4.24 | 4.59 |
| 100 | $5,518 | $640 | $2,003 | $931 | $127 | $4,666 | 4.87 | 19.81 | 5.95 | 16.78 | 6.35 | 6.78 |
| 250 | $8,668 | $1,023 | $3,467 | $1,418 | $198 | $7,569 | 7.65 | 31.65 | 10.30 | 25.56 | 9.92 | 11.00 |
| 475 | $11,431 | $1,374 | $4,813 | $1,872 | $258 | $10,219 | 10.09 | 42.53 | 14.30 | 33.75 | 12.94 | 14.85 |
| 500 | $11,671 | $1,405 | $4,932 | $1,913 | $263 | $10,446 | 10.30 | 43.49 | 14.65 | 34.49 | 13.19 | 15.18 |
| 1000 | $15,111 | $1,861 | $6,707 | $2,534 | $337 | $13,688 | 13.34 | 57.58 | 19.92 | 45.68 | 16.89 | 19.89 |
| **AAL** | **$938** | **$104** | **$163** | **$162** | **$13** | **$497** | **0.83** | **3.21** | **0.49** | **2.91** | **0.64** | **0.72** |


Tables above show the absolute and relative earthquake losses at country level and at regional level for the current exposure scenario (year 2020). Regional level refers to the aggregated results for the five Central Asia countries. Although the largest absolute losses are found for Uzbekistan, these values do not indicate that the largest earthquake risk in the region is in that 345 country. In relative terms (per mille - ‰), it can be seen from the same table that the Kyrgyz Republic and Tajikistan have larger losses due to a combination of the higher earthquake levels at locations with exposure concentrations, and earthquake vulnerability. From Tables 2 to 5, it can also be seen that this metric is additive, meaning that the regional AAL is the sum of the individual AAL's calculated for each of the five countries. However, the same additive property does not hold true for specific return period losses, meaning that the regional loss for a given return period is different (lower) than the sum of the 350 individual losses for that same return period calculated for each country.

When interpreting the absolute losses of the exposure representative of year 2080, it must be noted that it was developed only for the residential sector. Therefore, since the total exposed value is lower in the region for the 2080 case, the absolute losses





are lower too. This does not mean that risk will decrease in the future under the assumptions adopted in this study. Besides the

losses shown in the previous tables, earthquake losses can be further disaggregated by sector (into as many as included in the exposure databases). For instance, Figure 7 shows the normalized AAL at Oblast level in the study area for the commercial buildings.

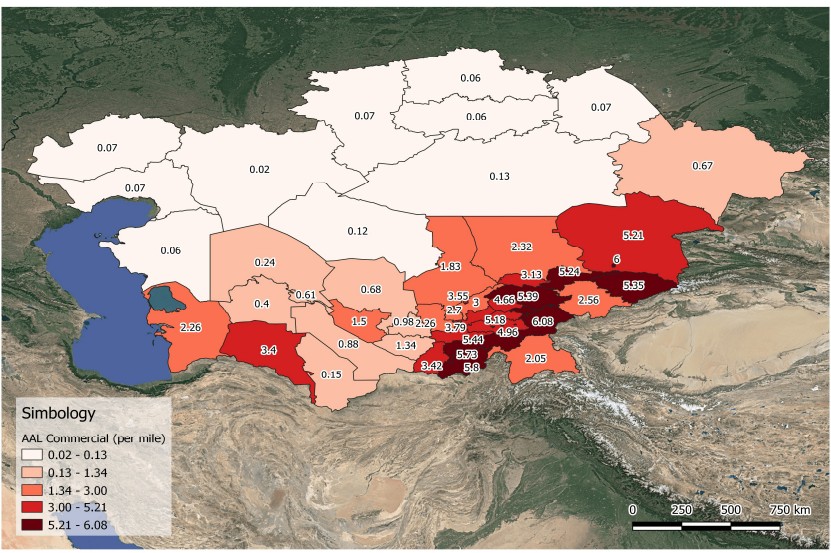

**Figure 7: Geographical distribution of the earthquake AAL (‰) by Oblast for the commercial buildings in Central Asia. Background**
**satellite image © Google Earth**

### 4.2 Scenario earthquake loss estimates

Some disaster risk management applications, as the development of emergency response plans, require, in addition to the probabilistic risk results, additional information such as scenario (or pseudo-deterministic) risk assessments, on which only one event is included in the risk equation. We denote this risk assessment approach as pseudo-deterministic since they are only

deterministic from the occurrence perspective (i.e., $F_A$ takes a value equal to 1.0 in Equation 4), given that the loss assessment is still probabilistic and the uncertainties associated to the hazard and vulnerability components are identified, quantified, and propagated. This approach yields results in terms of the expected losses (both economic and human), as well as allowing obtaining the probability density function (PDF) of the loss. In this project, five scenario analyses using the previously described approach were carried out, one for each city. The cities correspond to the capitals of each country, except for

Kazakhstan, where this scenario analysis was carried out for Almaty City because of the relatively low seismic hazard at Astana.

The criterion followed for choosing the critical scenarios was based on the disaggregation of the loss at an arbitrarily chosen return period, in this case set equal to 100 years. For this, the following steps were followed:


1. The LEC for the Oblasts where the cities of interest are located are used, and the loss with a 100-year return period is read as a reference. However, if a different return period is needed, all the required information is at the LECs.

2. Knowing he 100-year return period loss, from the year loss tables (YLT) different events that cause similar loss
amounts are identified. Depending on the seismotectonic and vulnerability characteristics, these events can have either, similar locations and magnitudes (e.g., all moderate magnitude at close distances to the city), or different



locations and magnitudes (e.g., some with moderate magnitude at close distances to the city and some with larger magnitudes but farther from the city).

3. From each of these events, the key parameters are identified, such as their magnitude, location, depth, and rupture characteristics (e.g., strike and dip). For each city, between 4 to 5 synthetic events causing similar losses than the one with the 100-year return period are identified.

    4. After that shortlisting, a single earthquake is chosen, and the single-event risk assessment is completed.

Table 6 shows the characteristics of the chosen earthquakes for the five analyzed cities. In general, all magnitudes are moderate except for the event in Turkmenistan where, based on the 100yr-loss disaggregation results, an event with $M_W > 7.0$ was selected. Figure 8 shows the modelled PGA for the Ashgabat event as an example.

**Table 6. Characteristics of the selected earthquake scenarios for the pseudo-deterministic earthquake risk assessment**

| Country | City | $M_W$ | Longitude (°) | Latitude (°) | Depth (km) |
|---|---|---|---|---|---|
| Kazakhstan | Almaty | 6.6 | 77.147 | 43.505 | 15 |
| Kyrgyzstan | Bishkek | 5.4 | 74.560 | 42.811 | 15 |
| Uzbekistan | Tashkent | 5.6 | 69.421 | 41.361 | 25 |
| Turkmenistan | Ashgabat | 7.1 | 58.477 | 38.404 | 15 |
| Tajikistan | Dushanbe | 5.8 | 68.725 | 38.712 | 15 |


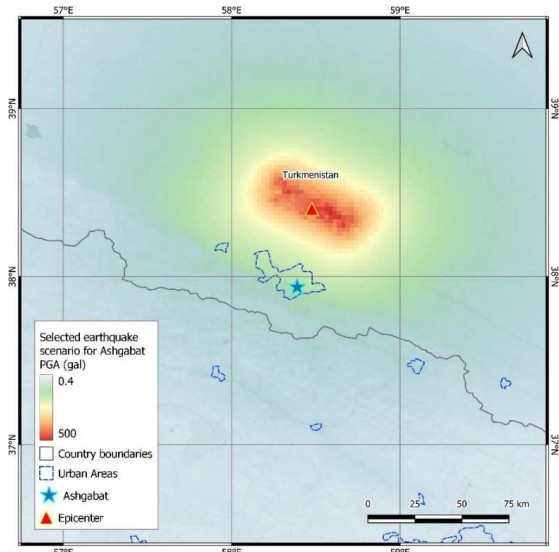

**Figure 8: Shakemap (PGA) for the deterministic event of Mw=7.1 at 50km of Ashgabat, Turkmenistan**

Table 7 shows the expected values for the economic losses and fatalities after performing the risk assessment for these five
events. These results include the economic losses modelled for all the sectors considered in the exposure database.





Table 7. Expected economic losses and fatalities for the pseudo-deterministic earthquake risk assessment

| Country | City | Loss (US M) | Fatalities |
|---|---|---|---|
| Kazakhstan | Almaty | 1,401 | 367 |
| Kyrgyzstan | Bishkek | 948 | 1,089 |
| Uzbekistan | Tashkent | 1,959 | 122 |
| Turkmenistan | Ashgabat | 381 | 159 |
| Tajikistan | Dushanbe | 325 | 434 |

## 5 Discussion and Conclusions

This paper presented the methodological framework for the development of a regionally consistent and fully probabilistic earthquake risk assessment for Central Asia, which yielded results in terms of loss exceedance curves (LEC), average annual losses (AAL) and specific return period losses, namely the commonly used risk metrics within any comprehensive disaster risk management strategy. Earthquake risk has been estimated considering four different exposure models, one representative for today's conditions, and three others for year 2080 but accounting for different shared socioeconomic pathways (SSP). The level of detail for all the components of this earthquake risk model is higher with respect to previous studies in the region. This refinement has been complemented too with the inclusion of additional sectors in the exposure databases that enabled the derivation of a more comprehensive overview of the earthquake risk level in the study area.

The earthquake losses calculated in this project allowed having an order of magnitude for the feasible losses up to the subnational level, which was possible because of the good enough resolution level used for the representation of both, hazard, and exposure. Also, by having used a regionally consistent approach, the results are directly comparable among them, besides having used consistent assumptions, modelling approaches and treatment of uncertainties which, considering the final objective of the study (i.e., the regional calculation of earthquake losses) is a key issue.

Additional to the earthquake risk, flood risk was also of interest within this project and because of that, a peril-agnostic and fully probabilistic risk assessment methodology was used. This was achieved by using the same representation of all the key risk components (i.e., hazard, exposure, and vulnerability) and a key benefit is having obtained the results in terms of the same risk metrics. At the same time, the exposure model disaggregated the assets into different sectors, therefore allowing the estimation of earthquake results for each of them and providing valuable information for policy and decision makers involved in different activities both, at subnational and national levels.

It must be noted too that the results of this earthquake risk assessment are intended to inform the World Bank's engagement in the support of regional and national disaster risk financing and insurance applications, which can include, for instance, traditional and parametric solutions for the structuring of a regional program. However, the level of detail of this study may not be sufficient to, alone, support the planning and design of specific risk management infrastructure, although clearly can be used to enable the World Bank to initiate a policy dialogue.

Central Asia lacks detailed analysis of historical emergency costs. Because of this, a methodology was proposed in the framework of the project (See Berny et al., 2023) considering all the relevant aspects identified after a literature review and taking as a reference the modeled earthquake loss.



A major limitation found during the development of this earthquake risk model was the lack of data to carry out a comprehensive validation and calibration process. A regional approach has been used to build the earthquake vulnerability model fin the five countries, peculiarities of each country have not been taken into account due to the lack of detailed data;
however, no significant differences within the region are expected and a calibration process was performed to match the reported losses available for historical events. Finally, and regarding the exposure projections to year 2080 of the residential buildings, it must be highlighted that catastrophe risk models always have associated levels of uncertainty when performed for the current hazard and exposure characteristics, that in this case tend to increase due to the uncertainties for this future timeline.
For this reason, although these results have been made available, they must be taken as indicative and only used for comparison purposes. Also, the normalized losses should be preferred over the absolute ones.

**Data availability**

Data and results of this study are available at: https://datacatalog.worldbank.org/search?q=sfrarr%20central&start=0&sort=

**Acknowledgments**

The Strengthening Financial Resilience and Accelerating Risk Reduction in Central Asia (SFRARR) project was funded by the European Union and was implemented by the World Bank. Authors are thankful to the World Bank Specialists, in particular Chyi-Yun and Stuart Fraser for their contributions and constructive criticism during the development of the earthquake risk model. Authors are also thankful to Dr. Sergey Tyagunov for the assistance and support in the finding and validation of local data.

**Author contribution**

MASG: conceptualization, validation, formal analysis, writing and editing. MOS: conceptualization, risk assessment methodology, software, validation, formal analysis, writing review and editing. BH, OG & CA conceptualization, software, formal analysis, and validation. EF, MK, PC and ZF: conceptualization, vulnerability model development, vulnerability methodology, formal analysis, validation, and calibration.

**Competing interests**

All authors declare that they do not have conflicts of interest of any type.

**Special issue statement**

This paper is to be considered in the Special Issue: Regionally consistent risk assessment for earthquakes and floods and selective landslide scenario analysis in Central Asia.

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

Report 240323-00.




**Appendix A: Earthquake risk results at Oblast level for the five countries in Central Asia**

**Table A1. Earthquake losses for different return periods and AAL for the residential sector in the Kyrgyz Republic. The first line shows the absolute loss (in US M) and the second line the normalized loss (‰)**

| Oblast | Modelled loss (US M) [Relative loss to replacement cost (‰)] | | | | | | | | | |
|---|---|---|---|---|---|---|---|---|---|---|
| | 5yrs | 10yrs | 25yrs | 50yrs | 100yrs | 250yrs | 475yrs | 500yrs | 1000yrs | AAL |
| Batken | $15 | $28 | $52 | $75 | $103 | $151 | $194 | $198 | $255 | $11 |
| | [4.966] | [9.509] | [17.701] | [25.577] | [35.217] | [51.744] | [66.614] | [67.924] | [87.39] | [3.845] |
| Chuy | $17.3 | $65.5 | $201.3 | $364.7 | $608.0 | $1,095.3 | $1,543.2 | $1,581.3 | $2,113.9 | $34.4 |
| | [1.555] | [5.889] | [18.091] | [32.768] | [54.63] | [98.413] | [138.661] | [142.085] | [189.94] | [3.088] |
| Jalal-Abad | $28.1 | $58.4 | $123.3 | $191.7 | $276.7 | $415.2 | $528.8 | $538.5 | $674.2 | $24.8 |
| | [4.139] | [8.605] | [18.146] | [28.218] | [40.744] | [61.127] | [77.85] | [79.273] | [99.254] | [3.658] |
| Naryn | $3.8 | $7.7 | $14.6 | $21.0 | $28.6 | $41.1 | $53.0 | $54.1 | $72.8 | $3.0 |
| | [2.816] | [5.639] | [10.738] | [15.481] | [21.042] | [30.301] | [39.035] | [39.856] | [53.607] | [2.224] |
| Osh | $34.80 | $75.00 | $169.40 | $275.80 | $415.20 | $661.40 | $892.90 | $914.10 | $1,247.80 | $37.10 |
| | [3.892] | [8.392] | [18.947] | [30.851] | [46.434] | [73.977] | [99.867] | [102.239] | [139.562] | [4.15] |
| Talas | $1.0 | $5.7 | $20.9 | $38.9 | $62.9 | $103.9 | $138.0 | $140.9 | $181.9 | $3.2 |
| | [0.86] | [4.799] | [17.646] | [32.903] | [53.175] | [87.812] | [116.65] | [119.111] | [153.754] | [2.692] |
| Ysyk-Kol | $12.6 | $25.4 | $49.9 | $73.9 | $102.3 | $148.8 | $197.0 | $202.0 | $285.8 | $10.5 |
| | [4.785] | [9.637] | [18.944] | [28.02] | [38.787] | [56.455] | [74.728] | [76.61] | [108.391] | [3.997] |


**Table A2. Earthquake losses for different return periods and AAL for the residential sector in Tajikistan. The first line shows the absolute loss (in US M) and the second line the normalized loss (‰)**

| Oblast | Modelled loss (US M) [Relative loss to replacement cost (‰)] | | | | | | | | | |
|---|---|---|---|---|---|---|---|---|---|---|
| | 5yrs | 10yrs | 25yrs | 50yrs | 100yrs | 250yrs | 475yrs | 500yrs | 1000yrs | AAL |
| Badakhshan Autonomous Mountainous Region | $3.2 | $5.6 | $10.1 | $14.7 | $20.3 | $29.2 | $36.3 | $36.9 | $45.5 | $2.6 |
| | [2.589] | [4.617] | [8.321] | [12.046] | [16.626] | [23.91] | [29.781] | [30.276] | [37.303] | [2.127] |
| Dushanbe | $2.8 | $13.7 | $64.7 | $153.8 | $288.4 | $524.5 | $724.6 | $741.6 | $986.3 | $12.6 |
| | [0.549] | [2.705] | [12.751] | [30.293] | [56.809] | [103.315] | [142.72] | [146.081] | [194.273] | [2.472] |
| Khatlon Province | $104.5 | $184.7 | $328.9 | $469.9 | $644.4 | $945.5 | $1,219.6 | $1,244.0 | $1,607.5 | $81.3 |
| | [5.609] | [9.911] | [17.651] | [25.219] | [34.587] | [50.747] | [65.458] | [66.766] | [86.276] | [4.366] |
| Sughd Province | $42.6 | $95.3 | $212.4 | $344.6 | $520.1 | $832.4 | $1,121.8 | $1,148.1 | $1,564.0 | $41.7 |
| | [2.351] | [5.26] | [11.722] | [19.014] | [28.701] | [45.931] | [61.901] | [63.354] | [86.303] | [2.299] |
| Cities and Districts of the Republican Subordination | $54.2 | $105.0 | $228.2 | $371.0 | $561.9 | $893.3 | $1,177.6 | $1,202.0 | $1,565.7 | $55.9 |
| | [3.732] | [7.235] | [15.723] | [25.559] | [38.714] | [61.546] | [81.135] | [82.819] | [107.876] | [3.849] |

**Table A3. Earthquake losses for different return periods and AAL for the residential sector in Turkmenistan. The first line shows the absolute loss (in US M) and the second line the normalized loss (‰)**


| Oblast | Modelled loss (US M) [Relative loss to replacement cost (‰)] | | | | | | | | | |
|---|---|---|---|---|---|---|---|---|---|---|
| | 5yrs | 10yrs | 25yrs | 50yrs | 100yrs | 250yrs | 475yrs | 500yrs | 1000yrs | AAL |
| Ahal | $6.4 | $17.3 | $44.8 | $81.1 | $133.6 | $224.5 | $299.8 | $306.2 | $396.7 | $8.1 |
| | [1.166] | [3.141] | [8.121] | [14.696] | [24.199] | [40.658] | [54.301] | [55.461] | [71.841] | [1.473] |
| Balkan | $4.3 | $8.8 | $17.1 | $25.3 | $35.5 | $52.7 | $67.7 | $69.0 | $87.8 | $3.4 |
| | [2.06] | [4.234] | [8.274] | [12.238] | [17.158] | [25.488] | [32.723] | [33.35] | [42.452] | [1.656] |
| Tashauz | $0.00 | $0.00 | $0.10 | $1.20 | $8.30 | $41.60 | $81.30 | $84.90 | $142.10 | $0.60 |
| | [0.0] | [0.0] | [0.026] | [0.367] | [2.622] | [13.09] | [25.547] | [26.698] | [44.684] | [0.204] |
| Chardzhou | $0.7 | $3.6 | $13.7 | $27.9 | $48.5 | $87.5 | $122.8 | $125.9 | $170.4 | $2.4 |
| | [0.174] | [0.872] | [3.331] | [6.783] | [11.802] | [21.294] | [29.876] | [30.625] | [41.47] | [0.58] |
| Mary | $0.0 | $0.0 | $0.3 | $1.2 | $4.7 | $18.2 | $42.3 | $45.2 | $101.2 | $0.4 |
| | [0.0] | [0.002] | [0.045] | [0.223] | [0.846] | [3.269] | [7.616] | [8.134] | [18.221] | [0.079] |

**Table A4. Earthquake losses for different return periods and AAL for the residential sector in Kazakhstan. The first line shows the absolute loss (in US M) and the second line the normalized loss (‰)**



| Oblast | Modelled loss (US M) | | | | | | | | | |
|---|---|---|---|---|---|---|---|---|---|---|
| | [Relative loss to replacement cost (‰)] | | | | | | | | | |
| | 5yrs | 10yrs | 25yrs | 50yrs | 100yrs | 250yrs | 475yrs | 500yrs | 1000yrs | AAL |
| Almatinskaya | $72.0 | $181.0 | $482.1 | $852.6 | $1,324.9 | $2,064.8 | $2,684.2 | $2,738.6 | $3,560.5 | $92.4 |
| | [1.168] | [2.937] | [7.824] | [13.835] | [21.5] | [33.508] | [43.559] | [44.441] | [57.78] | [1.499] |
| Almaty City Area | $0.7 | $12.0 | $129.6 | $439.4 | $1,007.0 | $2,123.0 | $3,146.4 | $3,236.3 | $4,541.5 | $38.8 |
| | [0.027] | [0.484] | [5.241] | [17.763] | [40.707] | [85.825] | [127.196] | [130.83] | [183.593] | [1.57] |
| Akmolinskaya | $0.00 | $0.00 | $0.00 | $0.00 | $1.00 | $5.80 | $14.30 | $15.50 | $69.40 | $0.40 |
| | [0.0] | [0.0] | [0.0] | [0.001] | [0.059] | [0.323] | [0.803] | [0.869] | [3.9] | [0.022] |
| Atyrauskaya | $0.0 | $0.0 | $0.0 | $0.1 | $1.1 | $6.3 | $14.2 | $15.0 | $31.4 | $0.3 |
| | [0.0] | [0.0] | [0.0] | [0.011] | [0.13] | [0.774] | [1.738] | [1.84] | [3.846] | [0.032] |
| Aktyubinskaya | $0.00 | $0.00 | $0.00 | $0.20 | $1.60 | $5.80 | $10.60 | $11.00 | $19.80 | $0.10 |
| | [0.0] | [0.0] | [0.0] | [0.027] | [0.173] | [0.643] | [1.165] | [1.215] | [2.184] | [0.011] |
| Vostochno-Kazachstanskaya | $4.3 | $10.3 | $25.8 | $47.5 | $85.3 | $186.6 | $299.0 | $309.3 | $479.2 | $6.2 |
| | [0.153] | [0.363] | [0.913] | [1.68] | [3.018] | [6.602] | [10.577] | [10.943] | [16.953] | [0.218] |
| Mangistauskaya | $0.00 | $0.00 | $0.00 | $0.10 | $0.60 | $3.60 | $9.60 | $10.30 | $28.50 | $0.20 |
| | [0.0] | [0.0] | [0.0] | [0.006] | [0.044] | [0.265] | [0.71] | [0.762] | [2.108] | [0.013] |
| Severo-Kazachstanskaya | $0.0 | $0.0 | $0.0 | $0.0 | $2.5 | $12.0 | $23.7 | $24.9 | $49.1 | $0.6 |
| | [0.0] | [0.0] | [0.0] | [0.001] | [0.078] | [0.376] | [0.741] | [0.779] | [1.538] | [0.02] |
| Pavlodarskaya | $0.00 | $0.00 | $0.00 | $0.40 | $2.70 | $11.20 | $26.10 | $27.70 | $54.30 | $0.30 |
| | [0.0] | [0.0] | [0.0] | [0.021] | [0.134] | [0.549] | [1.278] | [1.359] | [2.664] | [0.015] |
| Karagandinskaya | $0.0 | $0.0 | $1.0 | $3.9 | $10.8 | $37.2 | $78.1 | $82.8 | $195.4 | $1.2 |
| | [0.0] | [0.0] | [0.026] | [0.104] | [0.291] | [1.0] | [2.101] | [2.23] | [5.258] | [0.033] |
| Kyzylordinskaya | $0.00 | $0.00 | $0.00 | $0.30 | $1.80 | $20.60 | $53.50 | $56.80 | $117.90 | $1.00 |
| | [0.0] | [0.0] | [0.0] | [0.011] | [0.071] | [0.789] | [2.047] | [2.173] | [4.513] | [0.036] |
| Kustanayskaya | $0.0 | $0.0 | $0.0 | $0.3 | $3.4 | $14.2 | $27.4 | $28.8 | $54.3 | $0.4 |
| | [0.0] | [0.0] | [0.0] | [0.018] | [0.174] | [0.722] | [1.396] | [1.467] | [2.766] | [0.019] |
| Turkistan | $23.30 | $70.00 | $178.90 | $302.10 | $473.50 | $827.80 | $1,246.00 | $1,287.80 | $1,955.90 | $31.60 |
| | [0.644] | [1.936] | [4.949] | [8.354] | [13.095] | [22.892] | [34.459] | [35.614] | [54.09] | [0.873] |
| Zapadno-Kazachstanskaya | $0.0 | $0.0 | $0.0 | $0.3 | $3.4 | $8.7 | $13.9 | $14.4 | $23.0 | $0.3 |
| | [0.0] | [0.0] | [0.0] | [0.041] | [0.474] | [1.224] | [1.947] | [2.016] | [3.222] | [0.036] |
| Jambylslkaya | $16.5 | $39.2 | $85.3 | $135.6 | $208.3 | $373.3 | $562.0 | $579.8 | $857.8 | $16.4 |
| | [1.139] | [2.71] | [5.899] | [9.386] | [14.416] | [25.829] | [38.887] | [40.123] | [59.356] | [1.138] |






**Table A5. Earthquake losses for different return periods and AAL for the residential sector in Uzbekistan. The first line shows the absolute loss (in US M) and the second line the normalized loss (‰)**

| Oblast | Modelled loss (US M) [Relative loss to replacement cost (‰)] | | | | | | | | | |
|---|---|---|---|---|---|---|---|---|---|---|
| | 5yrs | 10yrs | 25yrs | 50yrs | 100yrs | 250yrs | 475yrs | 500yrs | 1000yrs | AAL |
| Andijan | $39.8 | $163.3 | $544.6 | $1,049.3 | $1,762.1 | $3,036.5 | $4,164.5 | $4,263.0 | $5,719.7 | $92.3 |
| | [0.885] | [3.633] | [12.12] | [23.352] | [39.215] | [67.576] | [92.68] | [94.872] | [127.29] | [2.055] |
| Bukhara | $3.1 | $24.2 | $90.0 | $214.2 | $537.1 | $1,336.9 | $2,004.4 | $2,059.3 | $2,812.3 | $24.8 |
| | [0.078] | [0.602] | [2.238] | [5.328] | [13.361] | [33.254] | [49.859] | [51.225] | [69.955] | [0.617] |
| Fergana | $74.20 | $294.30 | $911.80 | $1,658.30 | $2,644.10 | $4,349.10 | $5,891.70 | $6,029.70 | $8,123.50 | $147.10 |
| | [0.819] | [3.245] | [10.054] | [18.285] | [29.156] | [47.956] | [64.966] | [66.487] | [89.575] | [1.622] |
| Jizzakh | $3.7 | $24.9 | $149.8 | $451.5 | $1,051.0 | $2,245.1 | $3,290.7 | $3,380.6 | $4,680.1 | $42.8 |
| | [0.058] | [0.392] | [2.356] | [7.102] | [16.532] | [35.315] | [51.762] | [53.176] | [73.617] | [0.673] |
| Khorezm | $0.00 | $0.00 | $2.70 | $23.40 | $95.80 | $386.60 | $762.10 | $796.30 | $1,298.30 | $6.10 |
| | [0.0] | [0.0] | [0.095] | [0.817] | [3.35] | [13.514] | [26.64] | [27.839] | [45.385] | [0.212] |
| Namangan | $30.9 | $124.5 | $453.7 | $926.1 | $1,625.3 | $2,874.0 | $3,936.7 | $4,028.1 | $5,345.5 | $80.7 |
| | [0.644] | [2.59] | [9.438] | [19.264] | [33.808] | [59.783] | [81.889] | [83.79] | [111.194] | [1.678] |
| Navoiy | $0.1 | $1.8 | $9.7 | $25.2 | $67.5 | $219.2 | $432.7 | $452.8 | $733.8 | $4.1 |
| | [0.007] | [0.087] | [0.467] | [1.208] | [3.237] | [10.512] | [20.753] | [21.717] | [35.191] | [0.199] |
| Kashkadarya | $1.4 | $16.2 | $122.3 | $349.2 | $748.1 | $1,507.8 | $2,178.8 | $2,238.2 | $3,124.8 | $30.0 |
| | [0.026] | [0.304] | [2.293] | [6.547] | [14.027] | [28.272] | [40.855] | [41.968] | [58.592] | [0.563] |
| Karakalpakstan | $0.0 | $0.0 | $2.8 | $14.7 | $43.2 | $138.2 | $261.0 | $272.9 | $453.8 | $2.4 |
| | [0.0] | [0.0] | [0.084] | [0.445] | [1.309] | [4.19] | [7.909] | [8.27] | [13.753] | [0.074] |
| Samarkand | $0.2 | $5.2 | $69.7 | $265.8 | $751.2 | $2,041.1 | $3,427.3 | $3,555.4 | $5,560.5 | $35.4 |
| | [0.002] | [0.049] | [0.657] | [2.508] | [7.087] | [19.255] | [32.333] | [33.542] | [52.457] | [0.334] |
| Sirdarya | $3.4 | $29.1 | $193.3 | $504.1 | $1,012.0 | $1,981.7 | $2,842.1 | $2,916.7 | $3,995.3 | $41.8 |
| | [0.076] | [0.659] | [4.37] | [11.395] | [22.879] | [44.8] | [64.25] | [65.936] | [90.321] | [0.945] |
| Surkhandarya | $14.2 | $52.0 | $150.3 | $261.0 | $399.7 | $622.3 | $807.0 | $823.2 | $1,064.6 | $23.6 |
| | [0.728] | [2.666] | [7.715] | [13.393] | [20.511] | [31.936] | [41.415] | [42.242] | [54.629] | [1.209] |
| Tashkent | $73.8 | $301.2 | $938.9 | $1,758.2 | $2,943.0 | $5,100.8 | $7,021.8 | $7,189.4 | $9,627.8 | $158.1 |
| | [0.568] | [2.317] | [7.223] | [13.525] | [22.638] | [39.237] | [54.014] | [55.303] | [74.06] | [1.216] |
| Tashkent City | $0.7 | $14.2 | $158.8 | $576.8 | $1,547.0 | $3,956.9 | $6,445.9 | $6,664.5 | $9,668.5 | $66.0 |
| | [0.014] | [0.282] | [3.16] | [11.478] | [30.784] | [78.741] | [128.27] | [132.621] | [192.399] | [1.314] |
