# Peer review of "Development of a regionally consistent and fully probabilistic earthquake risk model for Central Asia"

_Natural Hazards and Earth System Sciences, 2023_

## Author Comment (AC1)

**Reviewer 1**

The paper is consistent with the title and the declared objectives. Nevertheless, improvements could be made to increase the study understanding, through a deeper illustration of the methodology in some parts.

In the revised version of the manuscript, additional details about the risk assessment methodology will be added to facilitate and increase the understanding of the study.

The text has too many references to other papers (often the ones in the same special issue), reducing a lot the descriptions of the different steps in the present one. In many cases the references are related to important parts of the methodology, this makes less fluent the reading and sometimes it reduces the reading comprehension. Obviously, in a single paper it's not possible to describe all the details of a complex study, but adding the essential aspects will improve it.

Now that all the papers regarding risk assessment for this special issue have been submitted, we have been able to review the main aspect of this comment and make sure that all the relevant details about the methodology are included in the revised version of the manuscript.

In the following some specific comments highlighting minor typos and integrating the general observations written before:

- Lines 85-90: it is better to remove these results from the introduction.
  - We have removed this paragraph that included the results of the study from the introduction.
- Lines 103-4: there is an error in the paragraph interruption.
  - In the revised version of the manuscript, we have amended this layout error.
- Line 121: change 2090 with 2080 as reported in the other paragraphs of the paper.
  - This correction has been made in the revised version of the manuscript.
- Line 123: change module with model.
  - This change has been implemented in the revised version of the manuscript.

- Lines 120-135: the descriptions of the exposure models are too short. It's clear that the complete description is in other papers of the special issue, but, as example, a table with the different classes, or a resumed description, could help the comprehension of this step of your methodology, in this paper. The same for paragraph 3.1. Moreover, in the paper there isn't a short description of the different "SSP" or a related reference

- Lines 110-118: something more has to be added about the hazard assessment

  - In the revised version of the manuscript, we have added more details about the earthquake hazard model developed in the framework of this project.

- In paragraph 3.1 the sentence "The functions collected were then harmonized and processed" doesn't permit to understand the procedure to obtain the curves

  - In the revised version of the manuscript, a more complete description of the followed procedure to obtain the earthquake vulnerability functions has been included.

- In paragraph 3.2 it's not very clear the calibration process, after the comparison of the "real" and calculated data

  - In the revised version of the manuscript, a more complete description of the followed procedure to calibrate the earthquake vulnerability functions, based on the comparison between the modelled and the reported losses, has been included.

- Line 307: change Figure 2 with Figure 3

  - This typo has been corrected in the revised version of the manuscript.

- Line 379: change "he" with "the"

  - This typo has been corrected in the revised version of the manuscript.

---

## Author Comment (AC2)

**Reviewer 2**

The paper describes results of regional probabilistic loss assessment for five countries in central Asia. It describes a case-study for the application of event-based seismic loss assessment at the regional level. One issue with the paper is the possibility to re-produce the results. Therefore, the method, the data used, and the validation should be described in sufficient details for others to be able to follow/reproduce. For instance, the paper does not offer much insight about how the stochastic catalogue is generated, how the vulnerability functions are developed, the characteristics of the exposure model, and how the projection into 2080 is developed (some socioeconomic pathways are mentioned, and the reader is referred to another work). Moreover, very little is shown in the paper by way of validation –mentioned also by the authors. In most cases, the authors refer to other works for details/validation. This approach reduces the autonomy of the paper and makes it harder to read and to follow.

We thank the reviewer for the comprehensive revision of our manuscript. We would like to note that this manuscript is part of a Special Issue on which different papers, that have to do with the development of one or more components to carry out the fully probabilistic and event-based risk assessment for earthquakes (this paper) and floods (the one by Coccia et al.), have been submitted. This paper presents the risk assessment methodology and what outputs of each component have been used. To avoid repetitions, we have preferred to cite and indicate where the readers can find additional and complete details regarding the development of each component (e.g., PSHA, exposure model, etc.).

In the revised version we will provide more details about the validation and calibration procedure, as well as take care of the specific comments made by the Reviewer which we agree allow improving the original manuscript.

The paper needs to specify the thematic datasets used, the sources of data, the resolutions, the spatial extent. This holds, especially, for the exposure datasets, the vulnerability models, seismic sources, the stochastic catalogues, the geological and geotechnical datasets, and the loss data from historical earthquakes. As a work showcasing the results of a regional risk assessment useful for decision making purposes, the results need more comprehensive validation (both at the local/global level). Have the authors thought of comparing with the results of the Global Earthquake Model, if available?

In the revised version of the manuscript, we will provide additional details about the spatial extend, resolution level, generation of the synthetic catalog.

Here are some more specific comments:

- Please describe what is mean by the "regionally-consistent" in the title?

  o An explanation of what is meant by the expression regionally-consistent will be included in the revised version of the manuscript. In a nutshell, it refers to an homogeneous approach to carry out the earthquake risk assessment, on which the same resolution level, assumptions and source data were used.

- Introduction and abstract: please specify the spatial extent for the 2bn AAL estimate. Is it all the five countries?

  o In the revised version of the manuscript, the specification of the 2bn AAL estimate will be provided. As the reviewer points out, it refers to the combined AAL for the five countries that are part of the study.

- Introduction, Line 85: It is not clear whether this part is related to the results of this paper or past studies. If these are findings of this paper, please move to the conclusions.

  o The values shown starting in L85 are the results of this study. We will make it clearer in the revised version of the manuscript.

- Line 100: what is meant by a long-term relationship? Please describe.

  o A description and more complete explanation about the meaning of a long-term relationship when dealing with earthquake risk assessment will be included in the revised version of the manuscript.

- Figure 1: The quality of the figure should be improved; the plots are too small and the labels cannot be seen.

  o The size of the fonts of labels and captions in all figures will be revised.

- Line 120: 2090 or 2080?

  o This value should be 2080 as correctly pointed out by the reviewer. The typo will be corrected in the revised version of the manuscript.

- Equation 4: please use a different notation like nu or lambda to indicate rate. F(.) is the notation for a cumulative distribution function (CDF), therefore it represents a probability and not a rate.

  o We prefer to keep the original notation, noting that the full explanation of all variables shown in Eq. 4 are included in the text.

- Line 200: This is the epistemic uncertainty in the prediction of the IM for a given event. It is estimated through a logic tree approach. please fix the wording.

- o We are referring to two types of uncertainties and the way they can be dealt with in a probabilistic earthquake hazard risk assessment framework. We believe the paragraph is correct and no modifications are needed.

- Figure 2: how these curves are derived? no explanation is provided. If they are derived based on literature, provide the statistics, the reference papers, information about the consequence model(s) used, information about the fragility curves, the number of damage states, etc.

  - o The revised version of the manuscript will include additional details about the development (and calibration) of the earthquake vulnerability functions.

- Figure 3: how this calibration is done? It seems that in some cases the difference with observed values has even increased after the calibration. Please describe the rationale for this calibration briefly.

  - o The revised version of the manuscript will include a more complete description of the vulnerability calibration procedure.

- Line 350: "However, the same additive property does not hold true for specific return period losses, meaning that the regional loss for a given return period is different (lower) than the sum of the individual losses for that same return period calculated for each country." Why is lower? Please explain.

  - o In the revised version of the manuscript, we will include more details about the risk metrics, to make it clear why the regional losses for a given return period is always lower than the sum of the individual country losses.

- Line 389: what is this shortlisting representing? how it done? what are the criteria?

  - o Since an event-based earthquake risk assessment was carried out, there are more than one synthetic earthquake that cause similar losses to that one with a 100-year return period. To that subset of possible synthetic earthquakes is that the shortlist is referring to. In the revised version of the manuscript this will be explained in more detail.

- Table 7: If these scenarios represent a 100 year return period, say so specifically.

  - o Yes, these scenarios are representative of a 100-year return period loss. In the revised version of the manuscript it will be made more explicit to avoid misunderstandings.

- Section 4.2: Scenario earthquake loss estimates. Perhaps, instead of calling them pseudo-deterministic, they could be referred to as scenario-based loss assessment. Then the authors could explain that the method is not fully deterministic.

    - Although we understand the suggestion of the reviewer, we prefer to maintain the original name for the analyses shown in Section 4.2, mainly to avoid misunderstandings with the event-based risk assessment methodology (which itself is fully probabilistic).

---

## Referee Report (RR1)

Second report

The authors have addressed my comments below reasonably. I have no further comments.

First report

The paper describes results of regional probabilistic loss assessment for five countries in central Asia. It describes a case-study for the application of event-based seismic loss assessment at the regional level. One issue with the paper is the possibility to re-produce the results. Therefore, the method, the data used, and the validation should be described in sufficient details for others to be able to follow/reproduce. For instance, the paper does not offer much insight about how the stochastic catalogue is generated, how the vulnerability functions are developed, the characteristics of the exposure model, and how the projection into 2080 is developed (some socioeconomic pathways are mentioned, and the reader is referred to another work). Moreover, very little is shown in the paper by way of validation – mentioned also by the authors. In most cases, the authors refer to other works for details/validation. This approach reduces the autonomy of the paper and makes it harder to read and to follow.

The paper needs to specify the thematic datasets used, the sources of data, the resolutions, the spatial extent. This holds, especially, for the exposure datasets, the vulnerability models, seismic sources, the stochastic catalogues, the geological and geotechnical datasets, and the loss data from historical earthquakes. As a work showcasing the results of a regional risk assessment useful for decision making purposes, the results need more comprehensive validation (both at the local/global level). Have the authors thought of comparing with the results of the Global Earthquake Model, if available?

Here are some more specific comments:

- Please describe what is mean by the "regionally-consistent" in the title?
- Introduction and abstract: please specify the spatial extent for the 2bn AAL estimate. Is it all the five countries?
- Introduction, Line 85: It is not clear whether this part is related to the results of this paper or past studies. If these are findings of this paper, please move to the conclusions.
- Line 100: what is meant by a long-term relationship? Please describe.
- Figure 1: The quality of the figure should be improved; the plots are too small and the labels cannot be seen.
- Line 120: 2090 or 2080?
- Equation 4: please use a different notation like nu or lambda to indicate rate. F(.) is the notation for a cumulative distribution function (CDF), therefore it represents a probability and not a rate.
- Line 200: This is the epistemic uncertainty in the prediction of the IM for a given event. It is estimated through a logic tree approach. please fix the wording

- Figure 2: how these curves are derived? no explanation is provided. If they are derived based on literature, provide the statistics, the reference papers, information about the consequence model(s) used, information about the fragility curves, the number of damage states, etc.
- Figure 3: how this calibration is done? It seems that in some cases the difference with observed values has even increased after the calibration. Please describe the rationale for this calibration briefly.
- Line 350: "However, the same additive property does not hold true for specific return period losses, meaning that the regional loss for a given return period is different (lower) than the sum of the individual losses for that same return period calculated for each country." Why is lower? Please explain.
- Line 389: what is this shortlisting representing? how it done? what are the criteria?
- Table 7: If these scenarios represent a 100 year return period, say so specifically.
- Section 4.2: Scenario earthquake loss estimates. Perhaps, instead of calling them pseudo-deterministic, they could be referred to as scenario-based loss assessment. Then the authors could explain that the method is not fully deterministic.
-

---

## Author Response (AR2)

**Reviewer 3**

**General**

The paper presents the approach and the results of a fully probabilistic earthquake loss model for Central Asia which was developed to respond to the need of a consistent, comparable and harmonized methodology across different perils in a multi-risk perspective (hazard and floods) and different regions. Such loss model is intended to provide a scientifically sound basis – albeit preliminary - for stakeholders dialogues for disaster risk mitigation policies. The methodological framework illustrated in the paper is based on an event-based risk assessment approach accounting for a full treatment of the uncertainty and for state-of-the-art approaches for the characterization of the seismic hazard, vulnerability and exposure components of the loss model. The efforts made by the authors to respond to the reviewers' remarks are acknowledged and clearly an improved version of the manuscript has been produced. Nevertheless, I believe that a few points still need to be clarified and explained in more detail in the manuscript, as commented below.

We thank the reviewer for this time and insightful comments that have allowed improving this manuscript. We are providing in blue a response to each comment, as well as an annotated version of the manuscript which shows how these have been addressed.

**Remarks**

1)For the seismic risk, a unique ground motion intensity measure (IM), i.e. Peak Ground Acceleration, is adopted regardless of the considered asset (buildings or infrastructures). I understand the reasons behind such a choice, especially in a context like the one in Central Asia with limited information on existing fragility and vulnerability curves. However, the choice of an efficient/sufficient/practical IM is crucial in a seismic risk model and it is widely recognized that PGA may not be optimal as an efficient predictor for damage and, eventually, losses, especially for some building typologies. Such aspects shall be commented by the authors. The adoption of a inefficient ground motion IM may be another reasons for the poor comparison with the reported losses (see Section 3.2).

We thank the Reviewer for this comment. We acknowledge it is a legitimate concern and we agree that choice of IM is a crucial point in any earthquake risk assessment. Nonetheless, as also fairly mentioned by the Reviewer, there is

limited information or even functions available for the region under study while most of the ones that are available are provided as a function of PGA. We are aware of this limitation, but it was a choice beyond the preferences of the authors. To address this comment we have added a sentence in section 3.1 of the revised version of the manuscript.

Furthermore, I noted that, in some parts of the manuscript, authors refer to "acceleration" (e.g. line 112, etc..) and "acceleration footprints" (line 278), while in other parts, as "PGA" (line 228, 250, etc..). I recommend to use a consistent notation and adopt "Peak ground acceleration" which is the correct one (maximum values of acceleration are indeed considered). For this reason, the label of the horizontal axis of Figure 2 shall be modified as "PGA".

In the revised version of the manuscript, we have made this consistent. (i.e., the first time it is called Peak Ground Acceleration (PGA) while in any later cross reference it is called PGA).

Figure 2: it is interesting and alarming that urban schools displays a vulnerability as low as unreinforced masonry.

This is a correct observation by the Reviewer. The school vulnerability function was developed based on the weighted average of a sub-set of vulnerability functions in the database. According to the exposure model, a large percentage of the buildings in Central Asia are built with unreinforced masonry or adobe. More specifically and based on a regional survey carried out as part of the project, all inspected schools are constituted by LBM (load-bearing masonry) or Precast concrete (80 and 20%, respectively). While in both urban and rural areas a large percentage of buildings was constructed between 1960 and 1990, urban areas have a larger fraction of modern schools (15% constructed after 2000) while rural areas have 15% of buildings constructed before 1960. Such large percentage of the adobe and unreinforced masonry evidently shifts the school buildings high up, close to these two building classes.

2) Referring to the social losses, does the methodology of this study allow to estimate only fatalities or other types of social losses (e.g. homeless)?

This study only considered the estimation of fatalities. Other types of social losses such as homeless are out of the scope of the study.

3) In my opinion, in Section 3.2, the approach adopted to calibrate the vulnerability functions by comparison with the observed losses during historical earthquakes is not explained with sufficient details. How do the authors adjust/modify the vulnerability functions to pass from the purple histogram to

the green histogram of Figure 3? Authors should provide details on this because it is central in the methodological framework.

In Section 3.2 of the revised version of the manuscript, we have added additional text to clarify this matter.

4) Among the interpretations provided by the authors to explain the differences between the observed and modelled losses for the 7 historical earthquakes (see section 3.2), might issues related to damage accumulation in seismic sequences be relevant?

We thank the Reviewer for this comment, and we agree with the appreciation. In section 3.2 of the revised manuscript, we have added additional text clarifying that the losses estimated in this study are associated only to the mainshock.

5) Figure 4. It is not clear whether, in the approach adopted by the authors specifically for historical earthquakes, the PGA footprint are provided by suitably selected empirical GMMs or, when available, they correspond to publicly available ShakeMaps. Of course, ground motion fields from GMM and from ShakeMap approach are different from a conceptual point of view.

We have added additional text on the revised version of the manuscript to clarify that these shakemaps correspond to PGA modelled after suitably selected empirical GMMs.

6) The aspects related to the choice of the spatial correlation model for ground motion intensity measure (PGA, herein) are not discussed in detail by the authors (see correlation coefficient in Eq. 3) and I believe that the discussion is relevant, as the choice of the correlation model may affect the results of spatially aggregated loss estimates.

In this study we have adopted the event-based risk assessment framework by Ordaz (2000) for which this discussion about the spatial correlation coefficient is explained in the original reference. We have added additional text in the revised version of the manuscript noting that the selection of the correlation model is likely to affect the spatially aggregated loss estimates.

7) I have some concerns on the outcomes of the pseudo-deterministic calculations for the 5 earthquake scenarios of Table 6. In particular, I see from Table 7 that the fatalities for the Mw5.4 Bishkek earthquake are of the order of 1000. It appears a rather large value for the relatively low Mw (which are the reasons for such high fatalities compared to other earthquake scenarios?) and, for validation purpose, this value should be compared with previous historical earthquakes with similar epicenter location and size to check the degree of

realism of such impact estimates. Also for the Tashkent earthquake of Mw5.6, the loss appears rather large and should be validated somewhat.

As explained in the manuscript, the selection of these five earthquakes was carried out based on the loss disaggregation, finding those that have estimated losses very close to that one of the 1000yr return period. This means that in some cases, ruptures were located closer or further away than the capital cities. In the case of Bishkek, the earthquake is very close to the city which combined with a relatively high vulnerability yields this number of estimated fatalities.

8) Figure 1 is scarcely readable and it should be enlarged/enhanced from a graphical point of view.

We consider that the relevant legends in Figure 1 are readable. However, if needed in the proofreading process these can be enlarged.